# Micromagnetic Study of the Dependence of Output Voltages and Magnetization Behaviors on Damping Constant, Frequency, and Wire Length for a Gigahertz Spin Rotation Sensor

**DOI:** 10.3390/s23052786

**Published:** 2023-03-03

**Authors:** Fumiko Akagi, Terumi Kaneko, Hirotada Kan, Yoshinobu Honkura, Shinpei Honkura

**Affiliations:** 1Department of Applied Physics, School of Advanced Engineering, Kogakuin University, Tokyo 163-8677, Japan; s419023@ns.kogakuin.ac.jp; 2Graduate School of Electrical Engineering and Electronic, Kogakuin University, Tokyo 163-8677, Japan; cm21011@ns.kogakuin.ac.jp; 3Magnedesign Co., Ltd., Nagoya 470-2414, Japan; yoshinobu.honkura@magnedesign.co.jp (Y.H.); shinpei.honkura@magnedesign.co.jp (S.H.)

**Keywords:** micromagnetic, gigahertz spin rotation sensor, damping constant, frequency, wire length

## Abstract

In this report, we studied the dependence of output voltage on the damping constant, the frequency of the pulse current, and the wire length of zero-magnetostriction CoFeBSi wires using multiphysics simulation considering eddy currents in micromagnetic simulations. The magnetization reversal mechanism in the wires was also investigated. As a result, we found that a high output voltage can be achieved with a damping constant of ≥0.03. We also found that the output voltage increases up to a pulse current of 3 GHz. The longer the wire length, the lower the external magnetic field at which the output voltage peaks. This is because the demagnetization field from the axial ends of the wire is weaker as the wire length is longer.

## 1. Introduction

Magnetic sensors detect the magnitude and direction of magnetic fields with high sensitivity without physical contact [1,2]. There are a wide variety of operating principles of magnetic sensors, and they are applied in many fields, such as the automobile, academic, industrial, and medical fields. Automobiles use more than 40 sensors, including angle sensors, speed sensors, and current sensors. They are important devices, indispensable for automatic driving. Magnetic sensors are also used in the biomagnetic field for measuring magnetoencephalograms and magnetocardiograms. Gyro sensors are used in wearable computers. Magneto resistance (MR) or tunnel MR sensors are used in magnetic hard disk devices and authenticity determination devices for bills and the like. Furthermore, in the aerospace field, gyro sensors are used for attitude control of spacecraft and artificial satellites. Figure 1 shows the relationship between the magnetic sensors and measurable magnetic flux densities (their sensitivities). The magnetic flux density required for biomagnetism and geomagnetism are 10^−14^–10^−9^ and 10^−6^–10^−5^ T, respectively, and that value is above those for industrial use. In the medical field, superconducting quantum interference device (SQUID) sensors are mainly used for biomagnetic measurements (magnetoencephalography, magnetocardiography) [3]. Fluxgate sensors are also used for magnetoencephalography with ~300 array sensors. Magnetoimpedance (MI) sensors and MR sensors are used as electronic compasses.

The SQUID sensor is an ultrasensitive magnetic sensor that utilizes the quantization phenomenon of superconducting magnetic flux and exhibits extremely high detection sensitivity on the order of 10^−15^ T. However, it requires a liquid helium cooling device and a magnetically shielded room, making it extremely large, expensive, and unportable. If we can develop a biomagnetic sensor that has sensitivity equivalent to that of a SQUID sensor that can operate at room temperature and is compact and inexpensive, it will lead to a significant increase in opportunities for the early detection of various diseases and health issues that make the most of its portability. One of the magnetic sensors proposed as an alternative to SQUID sensors is an MI sensor using amorphous wires (also called a giant magnetoimpedance (GMI) sensor) [4,5,6]. 

Mohri et al. [7] reported the history of MI (GMI) sensors. In 1981, UNITIKA Ltd. started producing amorphous alloy wires using a rapidly quenched spinning method in rotating water [8]. Afterward, various outstandingly sensitive magnetic effects such as Barkhausen [9,10], Matteucci [11], magnetoimpedance [6], and stress impedance [12,13] effects were found. In 1991, the GMI effect was found by Makhotokin et al. [4] in amorphous materials at a frequency of 2 MHz. In 1993, Mohri et al. developed a GMI sensor based on the GMI effect of an amorphous wire with a diameter of 30 μm, which provided high sensitivity at a frequency of 10 MHz based on the skin effect (see [2,5,6]). This strong skin effect is caused by the high circumferential permeability of the surface domain shell in the wire [14,15,16]. In 1999, Mohri et al. invented a coil-type GMI sensor with a pulse frequency of 20 MHz [17,18,19]. In 1998, Antonov et al. experimentally and theoretically studied the effect of the nondiagonal magnetoimpedance of amorphous wires with low magnetostriction [20]. In 2014, Zhukov et al. studied the effect of high frequency on GMI and off-diagonal GMI and reported that the GMI effect has a maximum output of ~200 MHz and decreases afterward [21]. More such studies have been recently reported [22,23,24].

The MI sensor currently has a detection sensitivity of 10^−12^–10^−2^ T, but it is necessary to improve the sensitivity for use in the field of biomagnetism. The MI sensor utilizes the phenomenon that when a high-frequency alternating current passes through an amorphous wire made of CoFeBSi, a soft magnetic material, an external magnetic field is applied only to the surface layer of the wire due to the skin effect, causing its impedance to change. Impedance is theoretically known to be proportional to 
√(f⋅μ(H))
 (*f*: frequency, *μ*: magnetic permeability, *H*: external magnetic field). In other words, the sensitivity of the MI sensor theoretically increases as the frequency increases, but until now, the pulse current was limited to the order of MHz [25,26,27] because the domain wall motion takes time, but the actual reason is not known. There are many studies on the magnetic domain structure using simulations, but they are often based on magnetostatic analysis [28,29,30,31,32]. Although some studies have been conducted using micromagnetic simulations to analyze magnetization behaviors, the skin effect (eddy current) has not been considered [31,32]. In 2015, Honkura et al. reported that when a GHz-order pulse current passes through a zero-magnetostriction CoFeBSi (having a weak negative saturation magnetostriction) amorphous wire magnetic alloy, high output voltage (high sensitivity) can be obtained. The reason is that only magnetization rotation in the wire surface contributes the output voltage. This effect is named the GSR effect, and a sensor that utilizes this effect is named a GSR sensor [33,34,35]. However, the actual magnetization reversal mechanism was not clear. Uehara et al. studied the mechanism of the GSR effect via computer analysis using the Landau–Lifshitz–Gilbert (LLG) equation assuming a spin vortex model in an amorphous wire with a diameter of 10 µm [35,36]. They also reported that the rise-time output voltage was ~50% larger than the fall-time output voltage. In addition, the relationship between magnetic field and coil voltage showed a sine function. These results were similar to their experimental results. These results suggest that the GSR effect could be based on spin rotation without the movement of magnetic walls. Honkura et al. developed a technology to produce microcoils and generated a microcoil on the ASIC surface, making a small-size GSR sensor possible [25,26]. Some prototypes of ASIC-based GSR sensors have been produced for applications in the automotive, body, gyrocompass, and medical fields. 

In a previous study, we investigated how the fall time of the pulse current affects the output peak voltages and the magnetization behavior inside an amorphous wire using a multiphysics simulation that includes the eddy current in the micromagnetics simulation [37,38]. At a fall time of 0.385 ns, only the rotations of magnetization in the outer circumference of the amorphous wire contributed to the output voltages, and at a fall time of 5.0 ns, the domain wall motions inside the wire also contributed to the output voltages. This difference was the reason for the increased output peak voltages as the fall time was shortened. 

In this report, we investigate the relationship between the frequency dependence of the output voltage and the damping constant using a multiphysics simulation that includes the eddy current in the micromagnetics simulation (Section 4). The damping constant is an important parameter, determining the high-frequency response. For example, Frommberger et al. reported the damping constant of FeCoSi derived from the exponential decay seen in the pulsed inductive microwave magnetometry curves [39]. Herein, we compared the output voltages at fall and rise times (Section 4). Honkura reported that reported that the output voltage has no hysteresis not only on the fall time but also on the rise time for the GSR sensor [27]. Additionally, as the output voltage at the rise time for the GMI sensor shows considerable hysteresis, only the output voltage at the fall time has been used thus far [26]. Therefore, the output voltage at the rise time is essential for developing a high output data rate (ODR)-type GSR sensor of over 1 MHz.

In addition, to miniaturize the sensor, the relationship between the axial length of the wire and the output voltage and the magnetization behavior of the wire (Section 5) were investigated.

## 2. Zero-Magnetostriction Amorphous Wire

### 2.1. Magnetic Domain Structure

In general, amorphous magnetic materials combine magnetic transition metals (Fe, Co, Ni, etc.) and semimetals (metalloids: B, Si, C, P), or magnetic transition metals and metals (Zr, Nb, etc.). They are produced in the form of ribbons, wires, or thin films by quenching, or they are produced in the form of films by methods such as sputtering, vapor deposition, and plating of the alloys of magnetic transition metals and rare-earth metals [2,9,10,11,12,13,14,15,40].

Figure 2 shows the magnetic domain structure generally considered for zero-magnetostriction amorphous wire [2,9,10,11,12,13,14,15,40,41]. The easy axis of the magnetization is induced in the circumferential direction of the wire due to the compressive stress generated in the surface layer of the wire during the ultra-quenching process in water. Furthermore, the circumferential magnetic anisotropy in the surface domain is strengthened by tension annealing after thinning by cold drawing after ultra-quenching. Since the inner domain of the amorphous wire cools more slowly than the surface layer, the shape magnetic anisotropy induces magnetic anisotropy along the wire axis, namely the easy axis of the magnetization is the longitudinal direction [2]. The magnetization of the middle domain between the surface and the inner domains is determined so that the total magnetic energy is minimized [2]. The magnetization of the middle domain is related to the distribution of the residual quenching stresses within the Co-rich amorphous wire [41]. The problem is that a demagnetization field is generated from the magnetic poles at both ends of the inner domain against an external magnetic field in the wire axial direction, reducing the effective magnetic field of the inner domain. In addition, minute magnetic domains are generated at both ends of the inner domain to reduce the magnetostatic energy, but because the magnetization changes due to discontinuous domain wall motion, it becomes a source of large magnetic noise. Hence, both ends of the inner domain of the wire are the sources of the demagnetization field and the magnetic noise. Therefore, to realize a compact and highly sensitive micromagnetic sensor, a sensing mechanism that does not use the inner domain of the amorphous wire is required.

### 2.2. Skin Effect

The skin effect can realize a micromagnetic sensor that “does not use the magnetization in the inner core of the amorphous wire” [2,17]. This is because the eddy current that suppresses the time change of the magnetic flux due to the circulating magnetic field in the conductor cancels the current at the inner core and promotes it at the surface. The depth from the surface at which the magnitude of the current is 1/e is called the skin depth and can be expressed using Equation (1).

(1)
$$\delta =\sqrt{2\rho /\omega \mu }$$

where 
δ
 is the skin depth, 
ρ
 is the electrical resistivity, 
ω
 is the angular velocity of the current, and 
μ
 is the magnetic permeability. Figure 3 shows permeability dependence of the magnetic field distribution in a cylindrical conductor calculated using Equation (2) [42]. The horizontal axis is the distance from the center of the cylindrical conductor to the circumference.

(2)
$$H\left(r\right)=\frac{I}{2\pi a}\;\frac{J_{1}\left(kr\right)}{J_{1}\left(ka\right)}\;,$$


*I* is the total current intensity flowing in the wire, *a* is the cylindrical conductor radius, *r* is the distance from the center of the cylindrical conductor, 
$J_{1}\left(kr\right)$
 or 
$J_{1}\left(ka\right)$
 is the Bessel function of the first kind (*k* = (1 − *i*)/*δ*). As shown in Figure 3, the larger *μ* is, the shallower the skin depth becomes, and the smaller the magnetic field, because the current does not flow in the center of the cylindrical conductor.

### 2.3. GSR Effect

When a pulse current of the order of GHz is passed along the axial direction of the amorphous wire, only the surface magnetization rotates in the circumferential direction at ultra-high speed due to the skin effect. The internal magnetic domains do not move because the domain wall motion cannot keep up with the rapidly changing magnetic field. This phenomenon was named the GSR effect [25,26,27]. Owing to this effect, the output voltage (sensitivity) increases up to a GHz-order pulse current. A GSR-based sensor winds a pickup coil around an amorphous wire and detects the change in magnetic flux density over time as an output voltage according to Faraday’s law of electromagnetic induction. 

## 3. Calculation Method

In this analysis, we used multiphysics simulation (inhouse program) considering eddy current calculation for micromagnetics simulation using the Landau–Lifshitz–Gilbert (LLG) equation [37,43,44,45,46,47].

### 3.1. Landau–Lifshitz–Gilbert Equation

The magnetization behavior in the amorphous wire was calculated via the LLG equation, as shown in Equation (3). This equation is numerically (Runge–Kutta method) solved using the finite difference method. When the magnetization vector points in a direction different from the effective magnetic field, it is in a nonequilibrium state; the magnetization vector points in the direction of the effective magnetic field while precessing.

(3)
$$\left(1+\alpha^{2}\right)d\vec{M}/dt=-\gamma \vec{M}\times {\vec{H}}_{\mathrm{eff}}-\gamma \alpha /M_{s}\cdot \vec{M}\times \left(\vec{M}\times {\vec{H}}_{\mathrm{eff}}\right)$$

where 
$\vec{M}$
 is the magnetization vector, 
${\vec{H}}_{\mathrm{eff}}$
 is the effective magnetic field vector (sum of the static magnetic field 
${\vec{H}}_{d}$
, exchange magnetic field 
${\vec{H}}_{\mathrm{ex}}$
, anisotropic magnetic field 
${\vec{H}}_{k}$
, and external magnetic field 
${\vec{H}}_{\mathrm{app}}$
), 
$M_{s}$
 is the saturation magnetization, *γ* is the ratio of the magnetic dipole moment to the angular momentum (gyromagnetic constant), and α is a damping constant representing the magnitude of damping action on the motion of magnetization. The first term on the right side of Equation (3) is an inertia term that acts on 
$\vec{M}$
 and expresses that the magnetization precesses about the effective magnetic field. The second term on the right side is a damping term that acts to orient 
$\vec{M}$
 in the direction of the effective magnetic field. In the case of only the first term, 
$\vec{M}$
 continues to precess with the effective magnetic field direction as the axis and does not converge. Due to the second term, a braking action works in the direction of the cross product of 
$\vec{M}$
 and 
$\vec{M\;}$
×
${\vec{\;H}}_{\mathrm{eff}}$
 (torque) that is the direction of the effective magnetic field, and 
$\vec{M}$
 converges in the direction of the effective magnetic field while spirally rotating. 
${\vec{H}}_{d}$
, 
${\vec{H}}_{\mathrm{ex}}$
, and 
${\vec{H}}_{k}$
 are calculated using Equations (4)–(8). The detailed calculation methods are presented in a previous study [46,47].

(4)
$${\vec{H}}_{d}=\sum_{\mathrm{all}\;\mathrm{cells}}\left[\begin{array}{ccc} S_{xx} & S_{xy} & S_{xz}\\ S_{yx} & S_{yy} & S_{yz}\\ S_{zx} & S_{zy} & S_{zz} \end{array}\right]\left[\begin{array}{c} M_{x}\\ M_{y}\\ M_{z} \end{array}\right]$$

where 
$\vec{M}$
 = (
$M_{x}$
*,*
$M_{y}$
*,*
$M_{z}$
) is the magnetization of a cuboid cell *m*, when the static magnetic field applies from the cell *m* to an observation cell *n*. The definitions of 
$S_{xx}$
 and 
$S_{xy}$
 in the tensor components in Equation (4) are shown in Equations (5) and (6), respectively.

(5)
$$S_{x,x}=\sum_{i,j,k}{\left(-1\right)}^{i+j+k}tan^{-1}\frac{\left(z_{ijk}^{\prime }-z\right)\left(y_{ijk}^{\prime }-y\right)}{\left(x_{ijk}^{\prime }-x\right)r_{ijk,n}}$$


(6)
$$S_{x,y}=-\sum_{i,j,k}{\left(-1\right)}^{i+j+k}log\left|\left(z_{ijk}^{\prime }-z\right)+r_{ijk,n}\right|$$

where (
x,y,z
) are the coordinates of the centroid of the cell *n*. 
$\left(x_{ijk}^{\prime },\;y_{ijk}^{\prime },\;z_{ijk}^{\prime }\right)$
 are the coordinates at the indices (*i, j, k*) of the eight vertices of the cell *m*. 
$r_{ijk,n}$
 is the distance between the vertex (*i, j, k*) and the centroid of cell *n*. Other tensor components are defined similarly.

(7)
$${\vec{H}}_{e}=\sum_{l=L,M,N\frac{2A}{M_{s}\delta^{2}}}\left[\begin{array}{c} m_{x,l+1}-m_{x,l-1}\\ m_{y,l+1}-m_{y,l-1}\\ m_{z,l+1}-m_{z,l-1} \end{array}\right],$$

where *A* (unit: J/m) is the exchange stiffness constant, 
δ
 is the distance between cells. 
$m_{\alpha ,l+1}$
 and 
$m_{\alpha ,l-1}$
 (
α
:* x,y,z, l*: 
L
*,*
M
*,*
N
) are the coordinates of the unit magnetization vectors of the nearest-neighbor cells of the observation cell (
L
*,*
M
*,*
N
)*,* where (
L
*,*
M
*,*
N
) are the indices.

(8)
$${\vec{H}}_{k}=\frac{2K_{u}}{M_{s}}\left(\vec{k}\cdot \vec{M}\right)\vec{k}$$

where *K_u_* (unit: J/m^3^) is the anisotropy constant, 
$\vec{k}$
 is a unit vector of the easy-axis direction.

### 3.2. Magnetic Field via Eddy Currents

The eddy current magnetic field vector 
${\vec{H}}_{\mathrm{eddy}}$
 was added to the effective magnetic field vector of Equation (3) [37,38,48,49]. The electric field (
${\vec{E}}_{\mathrm{eddy}}$
) is calculated via the relational expression (9) between the electric field 
$\vec{E}$
 and the magnetic flux density 
$\vec{B}$
 in Maxwell’s equations. Equation (10) is the integral form of Equation (9). The magnetic flux density 
$\vec{B}$
 is equal to 
$\mu_{0}\vec{H}+\vec{M}$
; 
$\vec{H}$
 is the sum of the eddy current magnetic field, static magnetic field, and external magnetic field (excluding the exchange and anisotropy fields). When a current flows through a conductor, a magnetic field via the current is generated around the conductor, and changes in this magnetic field generate eddy currents in a direction that hinders the current. The analytical expression of eddy current is obtained by solving the triple integral from the Biot–Savart law. The derivation process is shown below.

(9)
$$\nabla \times {\vec{E}}_{\mathrm{eddy}}=-\partial \vec{B}/\partial t=-\mu_{0}\partial \vec{H}/\partial t-\partial \vec{M}/\partial t.$$


(10)
$${\vec{E}}_{\mathrm{eddy}}=-\partial \left[\frac{1}{4\pi }\int \int \int \vec{B}\left({\vec{r}}^{\prime }\right)\times \left(\vec{r}-{\vec{r}}^{\prime }\right)/|\vec{r}-{\vec{r}}^{\prime }{|}^{3}d^{3}{\vec{r}}^{\prime }\right]/\partial t,$$

where 
$\vec{r}$
 is the position vector of the calculation point of the eddy current magnetic field, and 
${\vec{r}}^{\prime }$
 is the position vector of any point in the analysis domain.

(11)
$${\vec{J}}_{\mathrm{eddy}}=\sigma {\vec{E}}_{\mathrm{eddy}}\approx \nabla \times {\vec{H}}_{\mathrm{eddy}}$$


(12)
$${\vec{H}}_{\mathrm{eddy}}=\frac{1}{4\pi }\int \int \int {\vec{J}}_{\mathrm{eddy}}\left({\vec{r}}^{\prime }\right)\times \left(\vec{r}-{\vec{r}}^{\prime }\right)/|\vec{r}-{\vec{r}}^{\prime }{|}^{3}d^{3}{\vec{r}}^{\prime }$$

where *σ* is the conductivity, 
${\vec{J}}_{\mathrm{eddy}}$
 is the eddy current, and 
${\vec{H}}_{\mathrm{eddy}}$
 is the eddy current magnetic field. Because Equations (10) and (12) are the same triple integral, a common tensor matrix can be used for the calculations. Moreover, a two-dimensional fast Fourier transform was introduced to cope with the enormous computational cost. Furthermore, the following boundary conditions were given on the interface of the element.

(13)
$$\varepsilon_{0}div{\vec{E}}_{2}=\frac{2\partial {\vec{E}}_{\mathrm{eddy}}\;}{\partial n}\;.$$


(14)
$${\vec{J}}_{\mathrm{eddy}}=\sigma \left({\vec{E}}_{\mathrm{eddy}}+{\vec{E}}_{2}\right)\;,$$

where 
$\varepsilon_{0}$
 is the vacuum permittivity. Adding 
${\vec{E}}_{2}$
 cancels the surface normal component of 
${\vec{E}}_{\mathrm{eddy}}$
, which is the value calculated using (10) [48,49].

### 3.3. Magnetic Field Generated by Pulse Current and Output Voltage

The magnetic field generated by the pulse current was obtained using the following analytical formula, assuming that the current (*I*) uniformly flows through a cylindrical conductor with a radius of *a*.

(15)
$$H=Ir/2\pi a^{2}$$

where *r* is a distance from the center of the circle. The magnetic field is changed as per the pulsed current.

The output voltage is assumed to be the sum of the time variations of the axial component of the magnetization 
${\vec{M}}_{z}$
 of all the cells according to Faraday’s law, as shown in Equation (16). The output voltage (*V*) is normalized by the wire length.

(16)
$$V=-d\left(\sum_{\mathrm{all}\;\mathrm{cells}}{\vec{M}}_{z}\right)/dt.$$


## 4. Dependence of Output Voltage on Damping Constant

This section shows the dependence of the output voltage on the damping constant. 

### 4.1. Calculation Model

#### 4.1.1. Cylindrical Model

The amorphous wire has a cylindrical shape, but in this program, it was discretized with a rectangular parallelepiped cell. As shown in Figure 4, a cylindrical wire with a diameter of 10 µm and a length of 500 µm was discretized in a 0.2 µm × 0.2 µm × 50 µm cuboidal cell. Therefore, the number of cells in the *x* and *y* directions is 50, and the number of cells in the axial direction is 10. Here, the static magnetic fields normal to the surface in contact with the air were obtained by correcting (multiplied by a constant) the magnetization component to approximate the curved edges. The axis of easy magnetization of the amorphous wire was set to the z-axis direction, and the initial magnetization direction was set to the +z-axis direction.

#### 4.1.2. Pulse Current and External Magnetic Field

In a GSR sensor using an amorphous wire, when a pulse current flows in the z-axis direction of it, a magnetic field is generated in the circumferential direction, and the magnetization of the wire surface is oriented in the circumferential direction. The magnetization is oriented along the wire axis when no current is flowing. For the output voltage of the GSR sensor, the pickup coil detects the induced voltage generated by this magnetization change. Figure 5 shows the pulse current waveform and an example of the output voltage. We evaluated the output voltage at the fall time of the first pulse wave and the rise time of the second pulse wave, which are called fall-time detection and rise-time detection, respectively. The maximum pulse current was set to 0.39 A. The frequency *f* was defined as 1/(2*T*), where *T* is the rise and fall time of the pulse current. The interval between the first pulse wave and the second pulse wave was set to 10 ns, and the energizing time of the pulse current was 2 ns. The external magnetic field was 500 A/m.

#### 4.1.3. Magnetic Characteristics and Electric Resistivity

Table 1 shows the magnetic properties and electrical resistivity, which are typical values for amorphous FeCoSiB [7,37]. Herein, the damping constant was studied because FeCoSiB is an amorphous material with an exchange constant of 1.0 × 10^−11^ J/m.

### 4.2. Verification of Eddy Current Calculations

In this section, we determine whether the simulation program correctly incorporates the effects of eddy currents. Figure 6 shows the comparison between the output voltages at the rise time of the pulse current with and without the eddy current. The rise time was 0.8 ns for the pulse current. From this, the maximum output voltage without and with the eddy current is obtained as −0.341 V at 0.25 ns and −0.0861 V at 0.75 ns, respectively. Hence, the eddy current reduced the maximum output voltage and caused a large time delay. Therefore, we confirmed that the effect of eddy currents appeared in the calculation result.

### 4.3. Dependence of Output Voltage on Damping Constant

#### Output Voltage and Magnetization Behavior

Figure 7a,b shows the dependence of the rise-time output voltage and the fall-time output voltage on the damping constant, respectively. Both are the results of changing the frequency from 0.05 to 4 GHz. As shown, both output voltages at rise and fall times increased with the increasing frequency and peaked at 0.03–0.04 and slightly decreased after that. Therefore, a high output voltage can be obtained with a damping constant of ≥0.03. A previous study reported that the measured damping constant was 0.008 [39]. Based on this result, investigating materials or composition ratios with high damping constants may be required. We believe that it is necessary to measure and study the damping constant of zero magnetostrictive amorphous materials in the future. Furthermore, the rise-time output voltage was more than twice as high as the fall-time output voltage. We believe this is because the strong circumferential magnetic field from the pulsed current forces magnetization in the circumferential direction; therefore, the smaller the external magnetic field, the more easily the magnetization is oriented in the circumferential direction. As for the relationship with frequency, we confirmed that the fall-time output increases as the frequency increases but saturates at ~4.0 GHz. Similarly, the rise-time output voltage also increases as the frequency increases, and after exceeding 2 GHz, the increase becomes moderate and peaks at about 3 GHz. Therefore, we found that both rise- and fall-time output voltages increased up to the GHz order, which is the same result as the actual measurement.

## 5. Dependence of Output Voltage on Axial Length of Wire

This section presents the relationship between the axial length of the wire and the output voltage in order to reduce the size of the GSR sensor. We also investigated how the magnetization state changes with time when the axial length is changed.

### 5.1. Calculation Model

#### 5.1.1. Cylindrical Model

Figure 8 is a schematic diagram of the amorphous wire with a diameter of 12 µm. The axis length was changed from 200 to 500 μm. In this section, we set the easy axis on the surface of the wire to the in-plane direction and the easy axis in the inner wire to the axial direction, reflecting the effect of the compressive stress of the actual wire. The surface thickness of the wire was assumed to be 0.9 μm, which was identical to the ratio of the sensor diameter and the surface layer thickness in a previous study [37]. We also considered a smaller ratio than in another study (60 μm in diameter) [30]. In the future, we will study the relationship between the output voltage and surface thickness. The cell size for dividing the model was 0.2 μm × 0.2 μm in the in-plane direction, and the number of divisions in the axial direction was examined. Almost the same calculation results were obtained for 8 or more layers, but in this study, the model was divided into 16 layers. The initial magnetization direction was the +*z*-axis direction.

#### 5.1.2. Pulse Current and External Magnetic Field

Figure 9 shows the pulse current waveform and an example of the output voltage. In this study, the output voltages at the rise and fall times of the second wave of the pulse current were investigated. The maximum current value was 0.25 A; the rise time of the pulse current was 0.2 ns; and the fall time was 0.38 ns, which matches the actual measurements. The interval between the first pulse wave and the second pulse wave was set to 10 ns, and the energizing time of the pulse current was 4.5 ns.

#### 5.1.3. Magnetic Characteristics and Electric Resistivity

The magnetic characteristics were based on the values used for comparison with measurements in a previous study [37]. The magnetic characteristics inside the wire and electric resistivity of the amorphous wire were similar to that in Table 1. The wire surface was assumed to have a negative magnetic anisotropy to have anisotropy in the circumferential direction, where the anisotropy constant was −250 J/m^3^. The in-plane exchange constant was 1.0 × 10^−11^ J/m and that in the axial direction was 2.0 × 10^−11^ J/m to reproduce the measured results [37]. The damping constant was assumed to be the same as that of the previous study [37].

### 5.2. Experimental Methods

The ASIC-type GSR sensor was used to examine the effect of the wire length and the magnetic field on the output coil voltage. The measurement equipment and other conditions used in this study have been previously described in detail [26]. The GSR element used a wire with a composition of Co_50.7_Fe_8.1_B_13.3_Si_10.3_. In addition, the GSR element utilized glass-coated amorphous wires with a diameter of ~10 µm and the axial lengths of 160 μm and 900 μm. The GSR elements had a resistance of 4.32 and 11.927 Ω for the axial lengths of 160 μm and 900 μm, respectively. The difference in the resistances is due to not only the difference in the axial length but also the variation in the diameter of 15 ± 3 μm. The wire diameters are not the same, but the ratio of the axial length to the wire diameter is important. The wire with the axial length of 900 μm is longer than that with the axial length of 160 μm, even comparing the ratio of the axial length to the wire diameter. The number of coil turns was 14. The design values for the pulse current were a wire current pulse width of 5 ns, a rise time of 0.2 ns, and a fall time of 0.38 ns. The rise/fall times of the pulse current are the simulation values for circuit design. We believe that the reason for such a difference is the difference between the forced rotation of the spin at rise time and natural recovery process of the spin to the equilibrium state at the fall time.

### 5.3. Relationship between External Magnetic Field and Output Voltage for Each Axial Length

Figure 10 shows the relationship between the external magnetic field and the output voltage at axial lengths of 200–500 μm. As for the rise-time output voltage (Figure 10a), it increases with an increase in the external magnetic field for any axis length, reaches a peak value, and then decreases. The external magnetic field corresponding to the peak voltage was 4000, 2500, 1500, and 1000 A/m for the axis length of 200, 300, 400, and 500 μm, respectively. Alternatively, as the axial length increases, the rise-time output voltage reaches a peak value at a low external magnetic field. In addition, the peak rise-time voltages are almost the same, except for the slightly lower peak voltage value at 200 μm. As for the fall-time output voltage (Figure 10b), it also increases as the external magnetic field increases, reaches a peak value, and then decreases, regardless of the axial length. However, the external magnetic field of the peak fall-time voltage is higher than that of the rise-time output voltage. The external magnetic field corresponding to the peak voltage was 7000, 5000, 4000, and 3000 A/m for the axis length of 200, 300, 400, and 500 μm, respectively. As with the rise-time output voltage, as the axial length increases, the rise-time output voltage reaches a peak value at a low external magnetic field. In addition, the peak value became smaller as the axial length became shorter. Figure 11 shows the time dependence of magnetization in the *z*-axis (axial direction) in an external magnetic field of 500 A/m. From this, the shorter the axial length, the less the magnetization is oriented in the direction of the axial length during the 10 ns from the first wave to the second wave (the relaxation time). This is because the shorter the axial length, the more difficult it is for the magnetization to orient in the axial direction due to the demagnetization field at the end of the wire. Therefore, the shorter the axial length, the higher the external magnetic field required to direct the magnetization in the axial direction. Conversely, during the relaxation time between the first and second waves, the magnetization tends to be oriented in the axial direction because the longer the axis length is, the smaller the demagnetization field is. Therefore, the amount of magnetization in the circumferential direction increases at the rise time of the second wave, and the output voltage increases. Furthermore, at the fall time of the second wave, the longer the axial length of the wire because of the smaller demagnetization field, the more easily the magnetization is oriented in the axial direction, as in the first wave. Figure 12 shows the measured values of the relationship between the external magnetic field and the output voltage. From this, the external magnetic field, where the rise-time output voltage has a peak value, is 5171 and 795 A/m for the axis length of 160 and 900 μm, respectively. Therefore, the shorter the axial length, the higher the external magnetic field when the output voltage reaches the peak value. The same trend occurred for the fall-time output voltage. The external magnetic field, where the fall-time output voltage has a peak value, is 7160 and 1750 A/m for the axis length of 160 and 900 μm, respectively. Furthermore, the external magnetic field that reaches the peak value of the rise-time output voltage is lower than that of the fall-time output voltage. A similar trend was observed in the experimental results under other conditions of rise and fall times. These trends of the experiments also coincide with the simulation results shown in Figure 10.

## 6. Conclusions

We investigated the relationship between the frequency dependence of the output voltage and the damping constant using a multiphysics simulation that included the eddy current in the micromagnetic simulation. In addition, the relationship between the axial length of the wire and the output voltage and the magnetization behavior of the wire were investigated. The results are shown below.

A high output voltage can be obtained with a damping constant of ≥0.03.The output voltage increases up to a high-frequency current of 3 GHz.The shorter the axial length, the higher the external magnetic field when the output voltage reaches the peak value.The shorter the axial length, the stronger the demagnetization field at the end of the wire, making it difficult for the magnetization to return to the z-axis during the relaxation times of the first and second waves of the pulse current.The tendency of the dependence of the output voltage on the axial length agrees between the experimental and simulation results.

## Figures and Tables

**Figure 1 sensors-23-02786-f001:**
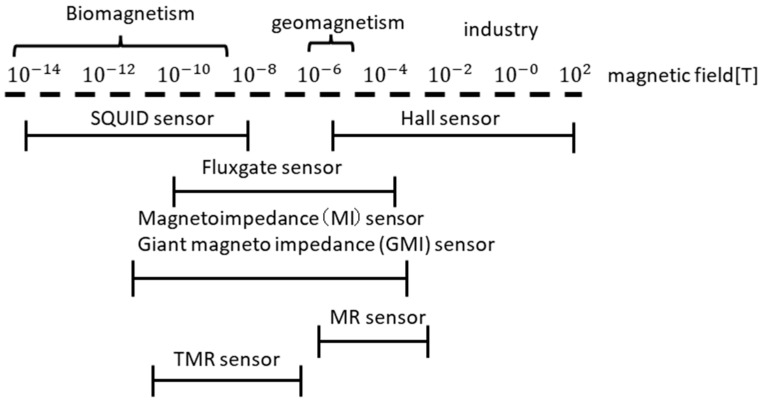
Relationship between magnetic sensors and measurable magnetic flux densities.

**Figure 2 sensors-23-02786-f002:**
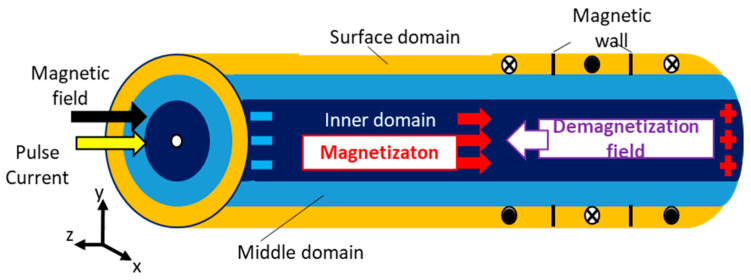
Magnetic domain structures in amorphous wire.

**Figure 3 sensors-23-02786-f003:**
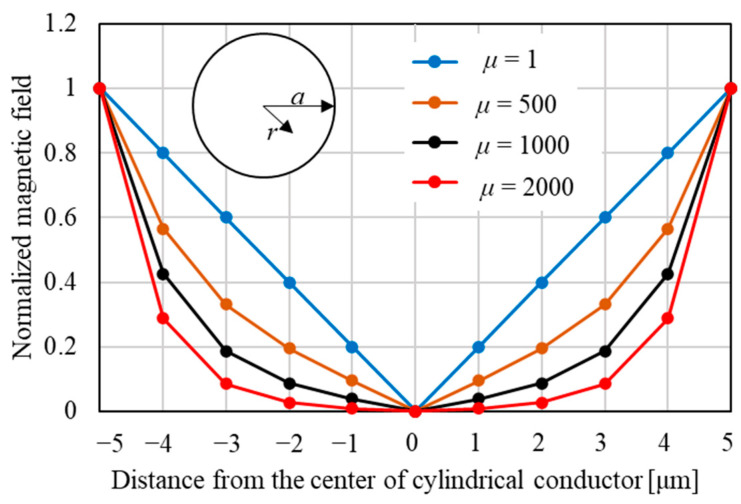
Permeability dependence of magnetic field distribution in cylindrical conductor.

**Figure 4 sensors-23-02786-f004:**
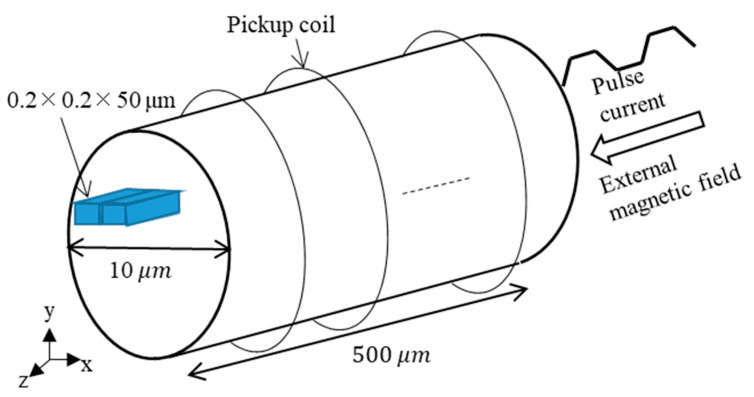
A cylindrical wire model with a diameter of 10 µm and a length of 500 µm.

**Figure 5 sensors-23-02786-f005:**
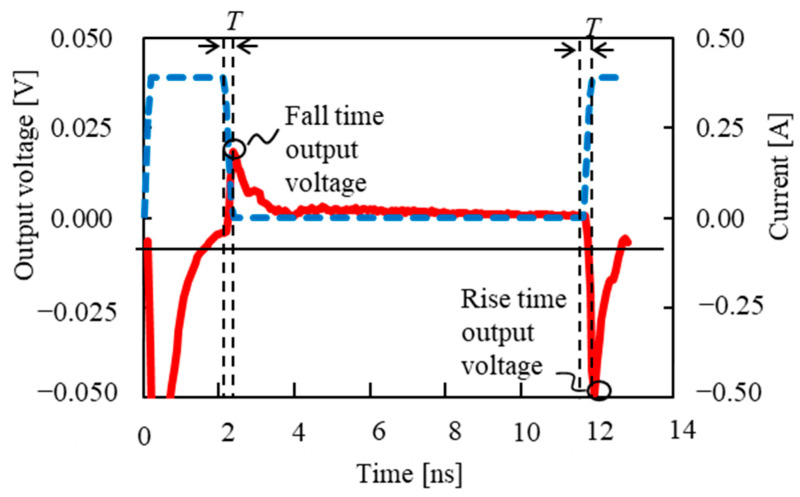
Pulse current waveform and an example output voltage.

**Figure 6 sensors-23-02786-f006:**
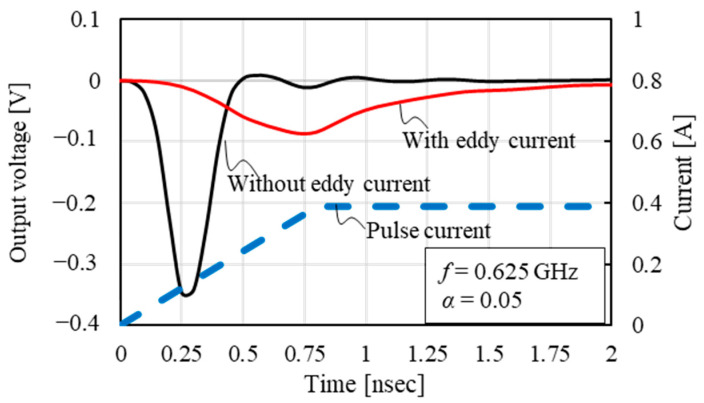
Comparison between output voltages at rise time of pulse current with and without eddy current.

**Figure 7 sensors-23-02786-f007:**
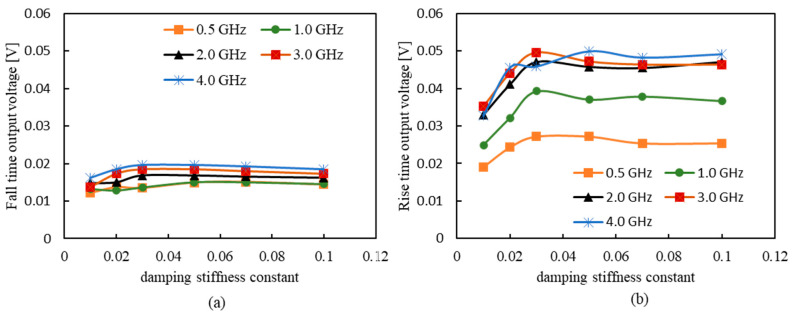
Damping constant dependence of output voltage: (**a**) rise-time output voltage; (**b**) fall-time output voltage.

**Figure 8 sensors-23-02786-f008:**
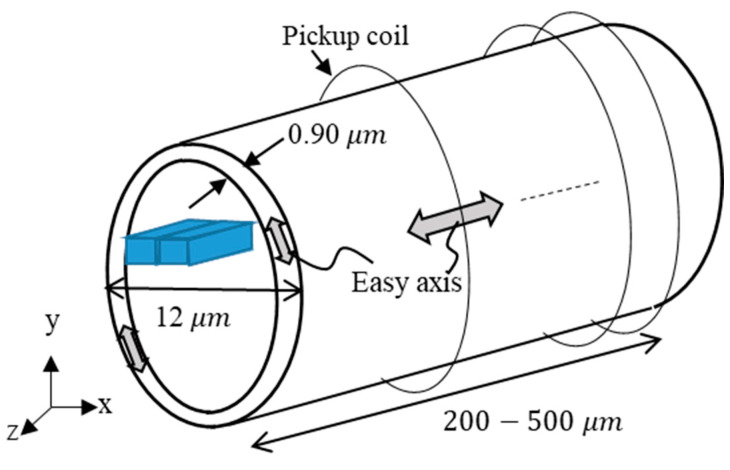
Schematic diagram of the amorphous wire for an axial length of 200–500 μm.

**Figure 9 sensors-23-02786-f009:**
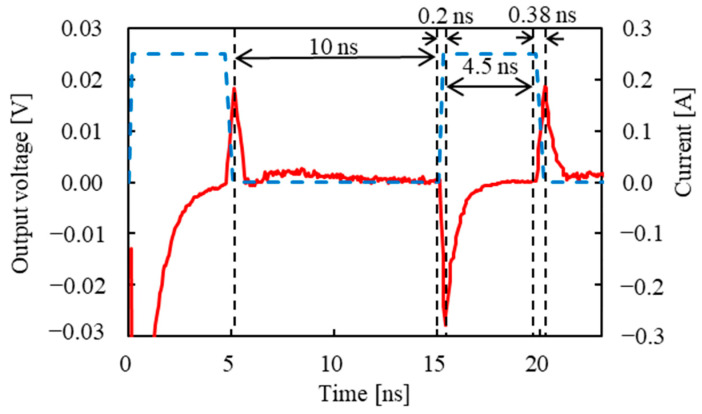
Pulse current waveform and an example output voltage.

**Figure 10 sensors-23-02786-f010:**
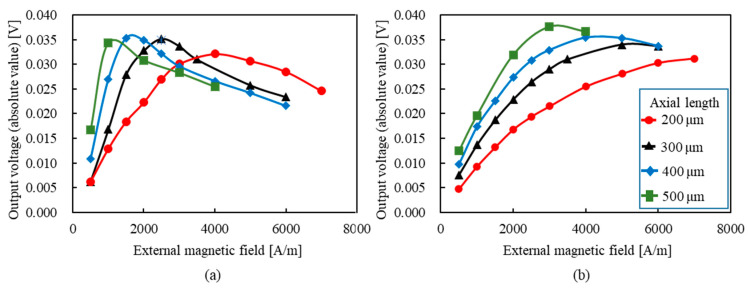
Relationship between the external magnetic field and the output voltage: (**a**) rise-time output voltage; (**b**) fall-time output voltage.

**Figure 11 sensors-23-02786-f011:**
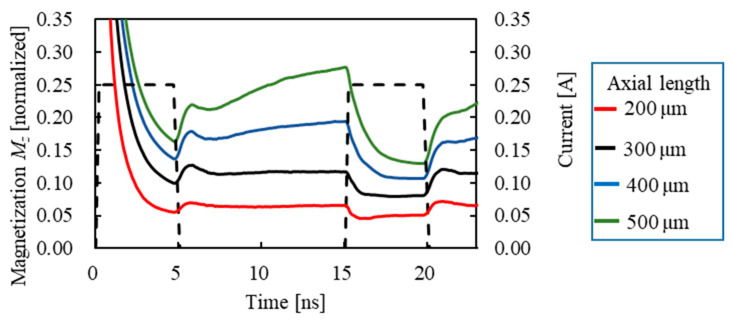
Time dependence of magnetization in the z-axis direction (axial direction).

**Figure 12 sensors-23-02786-f012:**
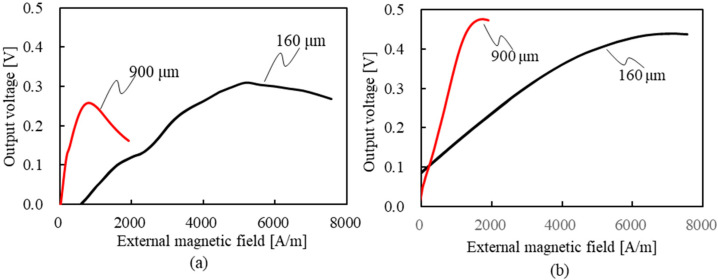
Measured values of the relationship between the external magnetic field and the output voltage: (**a**) rise-time output voltage; (**b**) fall-time output voltage.

**Table 1 sensors-23-02786-t001:** Magnetic characteristics and electric resistivity.

Parameter	Value
Anisotropy constant [J/m^3^]	250
Saturation magnetization [T]	1.0
Exchange stiffness constant [J/m]	1.0 × 10^−11^
Electric resistivity µΩ m	1.3

## Data Availability

Data sharing not applicable.

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
