# Peer review of "Micromagnetic Study of the Dependence of Output Voltages and Magnetization Behaviors on Damping Constant, Frequency, and Wire Length for a Gigahertz Spin Rotation Sensor"

_sensors, 2023, doi:10.3390/s23052786_

Round 1

Reviewer 1 Report

The authors studied dependence of output voltage on damping constant, frequency of pulse current, and wire length of zero-magnetostriction CoFeBSi wires using. It is helpful for the development of the gigahertz spin rotation sensor. However, there are also some problems in the manuscript. The necessary simulation analysis is not sufficient. The description of some figures in the manuscript is confused and inconsistent. These need to be clarified before being acceptable.

1.There is little description about GSR sensors in the introduction. It is better to introduce research actuality and specific application background of the GSR sensor.

2. Please give the detailed reason why the fall time output voltage is more than twice as high as the rise time output voltage in Figure. 7. Is it related to rising time and falling time?

3. In the part of cylindrical model, the surface thickness of the wire is 0.9 nm in line 302. While the thickness in Figure. 8 is 0.9 μm. If the thickness is different, dose it influence the follow-up study?

4. In the part of experimental method, the design values for the pulse current were a wire current pulse width of 5 ns, a rising time of 0.2 ns, and a falling time of 0.38 ns. Why are the rising time and falling time set differently? How is it done?

5. The description of Figure. 12 from line 371 to line 380 is confused and it is not consistent with the content of the Figure 12.

6. It is stated that the measurement result coincides with the simulation result in line 379. However, the analysis of the simulation results is not seen in this part. There is not figure or data table about the simulation results. How to prove the consistency?

Reviewer 2 Report

this is an interesting paper looking at using magnetostrictive wires as spin rotation sensors. it have the following comments:

 - please check the English there is the odd place it is not correct for example pg 1, line 29, rather than "It is", "They are" would be better

- pg 5 line 164 the sentence "When the magnetisation.." doesn't make sense, please consider rewriting it

- line 173 check that you mean to reference equation (2) in that paragraph

- line 229 "has a cylindrical shape but in this program it was discretized with a rectangular parallelepiped cell" - how are the curved edges resolved in the model, as they are important for the direction of the magnetisation. Does the "squareness" of the cells affect the overall behaviour of the wire

- the values in table 1 were they experimentally measured or take from literature, references should be provided for them

- normal practice is there is a space between the number and the unit, in places in the paper this does not occur such as lines 265, 266 etc

- line 284 you mention an actual measurement, can you either add a reference, or give the figure number here

- why were the rise and fall times investigated?

- line 330 methods is spelt wrong

- line 347 "falle" not sure what this means

 - start of the paragraph on line 371 is a repeat from the previous sentence, please combine

- line 396 can the damping constant values be experimentally explored?

- line 398 does not make sense

Reviewer 3 Report

In my opinion, this paper cannot be published in its present form.

First, it seems to have been written carelessly. Most probably, the authors have not checked with attention the final version that was sent to the journal. The text is full of mistakes, badly constructed sentences, orthography errors and even repeated paragraphs.

Second, I humbly understand that scientific articles must provide an increase of knowledge and present the findings in a way that can be followed (and reproduced, if it is desired) by the readers.

This is not the case with section 3 “Calculation method”, most of all with subsections 3.2 and 3.3. The presented calculation procedure is not new for this paper, and the authors cite reference [19] as a source (the title of this citation is missing in the reference list). For example, the steps for calculating H_eddy are sketched, but I have not been able to understand them. So, I went to reference [19], where it is better explained, but the key step of the calculation is also missing and pointed to a new reference (reference [17] of reference [19] of this paper). And that new reference is “Proc MMM, 2019” which I was completely unable to find. In definitive, the reader is not capable of understand (or reproduce) the calculation procedure.

I must say that I encountered the same situation is another reference of this paper (reference [14]), although this is not so relevant for the present investigation.

To illustrate another inconsistency, section 5.1.1 states that “the surface thickness of the wire was 0.9 nm [19].” This impossible number must be a mistake (in fact in Figure 8, the number is 0.9 µm), so I checked reference [19], where the number is 0.75 µm and again, there is no clue in either manuscript as why this number has been chosen and why it is different in both studies.

Finally, from the scientific point of view, I find that the main interesting part of the manuscript is the last section “Dependence of the output voltage on axial length of the wire”. The authors present there a rather interesting investigation, with sound results and good correlation between simulation and experiment.

However, I cannot see the interest of the previous section where the performance of the device is studied theoretically as a function of the damping parameter. The damping parameter is characteristic of the type of material used and cannot be changed experimentally without changing the material. Using different materials implies changing many other parameters in the model, so I don’t see the interest of this part of the investigation.

In definitive, in my opinion, the paper could be worth publishing if centered in the results of section 5, provided that the authors put interest in improving the writing of the manuscript, organizing the information and giving explanations clear enough to reproduce the results.

Reviewer 4 Report

The paper presents the results of a study of the output voltage of a GSR sensor based on amorphous wire from damping constant, frequency, and wire length. The manuscript is of interest to the scientific community. However, in the opinion of the Reviewer, it should be revised.

The most notable flaws in the manuscript are described below.

1. The magnetic structure of amorphous soft magnetic wires obtained by rapid quenching is determined by the distribution of residual quenching stresses. This has already been discussed previously, for example, in 10.1088/0022-3727/32/15/305 и 10.1088/0022-3727/29/4/001. The authors should discuss this in section 2.1. Moreover, we need to discuss the middle domain in more detail and add appropriate references to the literature. The authors experimentally study glass coated amorphous wires, so the features of their domain structure should be discussed in Section 2.1.

2. The reviewer recommends that authors ensure that they use the same quantity designations throughout the text. For example, line 169: “???? is the effective magnetic field vector (sum of static magnetic field, exchange magnetic field, anisotropic magnetic field, and external magnetic field)”; line 196: “? is the sum of the eddy current magnetic field, the static magnetic field and the external magnetic field” (is this the same as ?????); line 219: ? the magnetic field of the current. All this greatly complicates the reading of the manuscript.

3. Expression (10), which describes the dependence of the magnetic field on the radial coordinate, is true only for DC. Moreover, the authors have already introduced this dependence before (see expression (2)). Expression (2) is better suited for the case of AC. However, it should be borne in mind that (2) is valid only in the case of a homogeneous conductor, while amorphous wires have a non-uniform magnetic structure.

4. The authors pass current through a wire and pick up a signal from a coil wound around the wire. A similar scheme is used in the case of an off-diagonal magneto-impedance effect (off-diagonal MI). The off-diagonal MI theory is well developed (see, for example, 10.1016/S0304-8853(98)00114-0) and generally does not differ from the theoretical model presented in the manuscript. Therefore, the reviewer believes that it is necessary to pay attention to this effect.

5. What software package was used for the multiphysics simulation? There is no description for this.

6. The tables contain such parameters as “Anisotropy constant” and “Exchange stiffness constant”. These parameters were not previously introduced into the theoretical model (formula (3) and below). This deficiency needs to be corrected.

7. Part 5 discusses the dependence of the output voltage on the length of the wire. At the same time, the reviewer failed to find a description of how the dependence of the effective magnetic field on the wire length was specified.

8. The experiment is poorly described. Nothing is said about the experimental equipment.

9. It is not clear why a 160 µm wire has a resistance of 4.32 Ω, while a 900 µm wire has a resistance of 11.927 Ω. The lengths differ by a factor of 5.6, and the resistances differ by a factor of 2.7.

10. Experimental results and simulations are difficult to compare because there are no data on wires other than electrical resistance.

11. The experimental results look poor. For example, there is no variation in rising time and falling time. How are the authors going to vary the damping constant in the experiment?

12. The authors write: “A damping constant of about 0.03 to 0.04 provides the highest output voltage”. However, Figure 7 shows that at 0.03 the voltage reaches its maximum and then remains almost unchanged. Therefore, the conclusion of the authors looks unconvincing.

Minor disadvantages:

- Figure 1 is missing a letter in the word “Magnetoimpedance”.

- Tables 1 and 2 are similar in many ways. It is enough to leave one of them.

Round 2

Reviewer 1 Report

I have carefully reviewed the author's responses to my comments on the first manuscript and the revised paper, the authors have revised all issues in accordance with the comments, and I agree to the paper's publication on Sensors.

Reviewer 3 Report

The authors have made now an effort to make the manuscript comprehensible. It would have been better to have this version in first place.

The contents have not been modified substantially.

The calculation method in section 3 is explained in more detail but, sincerely, It seems to me still very difficult to reproduce with the information provided, in the paper or in the cited references.

The relevance of section 4, is slightly emphasized with a new sentence. Please, consider adding also the justification that was provided in the response letter about “how much the damping constant of the actual material is different from the value that can obtain high output voltage for future examination of materials and composition ratios”.

Apart from this,

-       correct the definition of SQUID in line 37 (superconducting)

-       use “Section” instead of “Chapter” in lines 113 and 125.

Revise the references. For instance: Reference 47 is incomplete. Is Reference 21 a valid citation? Revise journal abbreviations.

Reviewer 4 Report

The authors considered all comments.
